

# Effects of company and season on blood fluke (*Cardicola* spp.) infection in ranched Southern Bluefin Tuna: preliminary evidence infection has a negative effect on fish growth

Cecilia Power[1], Melissa Carabott[1], Maree Widdicombe[1], Lachlan Coff[1], Kirsten Rough[2], Barbara Nowak[1] and Nathan J. Bott[1]

[1] School of Science, Royal Melbourne Institute of Technology, Melbourne, Victoria, Australia
[2] Australian Southern Bluefin Tuna Industry Association, Port Lincoln, South Australia, Australia

Corresponding author
Nathan J. Bott,
nathan.bott@rmit.edu.au

## ABSTRACT

Aporocotylid blood flukes *Cardicola forsteri* and *C. orientalis* are an ongoing health concern for Southern Bluefin Tuna (SBT), *Thunnus maccoyii*, ranched in Australia. Therapeutic application of praziquantel (PZQ) has reduced SBT mortalities, however PZQ is not a residual treatment therefore reinfection can occur after the single treatment application. This study documents the epidemiology of *Cardicola* spp. infection in ranched SBT post treatment over three ranching seasons (2018, 2019 and 2021). Infection prevalence (percentage of SBT affected) and intensity (parasite load) was determined by adult fluke counts from heart, egg counts from gill filaments and the use of specific quantitative polymerase chain reaction (qPCR) for detection of *C. forsteri* and *C. orientalis* ITS-2 DNA in SBT hearts and gills. SBT Condition Index decreased as intensity of *Cardicola* spp. DNA in SBT gills increased, suggesting blood fluke infection had a negative effect on SBT growth (Spearman's r = −0.2426, d.f. = 138, p = 0.0041). Prevalence and intensity of infection indicated PZQ remained highly effective at controlling *Cardicola* spp. infection in ranched SBT, 10 years after PZQ administration began in this industry. Company A had the highest prevalence and intensity of *Cardicola* spp. infection in 2018, and Company G had the highest in 2019. No consistent pattern was seen in 2021. Overall, intensity of infection did not increase as ranching duration increased post treatment. Results from this study improve our knowledge of the biology of blood flukes and helps the SBT industry to modify or design new blood fluke management strategies to reduce health risks and improve performance of SBT.

## INTRODUCTION

Southern Bluefin Tuna (SBT), *Thunnus maccoyii* (Castelnau, 1872), is a major aquaculture species in Australia (*Steven, Dylewski & Curtotti, 2021*). Wild juvenile SBT (2–5 years of age) are caught in the Great Australian Bight and typically ranched for 2–7 months in open ocean pontoons off the coast of Port Lincoln, South Australia (*Ellis & Kiessling, 2016*).

Ranching was introduced in the 1990s to add quality and value to industry catch quotas, turning it into a high value fishery predominantly exported to the Japanese sashimi market (*Balli et al., 2016*). An ongoing challenge for aquaculture in open ocean pontoons is the exposure to pathogens (*Paladini et al., 2017*).

Blood flukes (*Cardicola* spp., Aporocotylidae) are important pathogens of bluefin tuna ranched and farmed across Australia, Asia and Europe (*Power et al., 2020*). In Australia, two species of *Cardicola* have been reported from SBT, *C. forsteri* and *C. orientalis* (*Cribb, Daintith & Munday, 2000*; *Shirakashi et al., 2013*). *Cardicola* spp. infects the circulatory system of SBT, with adult *C. forsteri* typically found in the heart and adult *C. orientalis* in the gills (*Colquitt, Munday & Daintith, 2001*; *Shirakashi et al., 2013*). *Cardicola* spp. eggs are released into the bloodstream and high severity of infection can cause blockages and lesions in the gills which can lead to mortalities (*Dennis, Landos & D'Antignana, 2011*; *Hayward et al., 2010*). The anthelmintic praziquantel (PZQ) has been used by the SBT industry since 2012 and reduced mortalities significantly (*Hardy-Smith et al., 2012*; *Power et al., 2019*). However, not all ranched SBT are PZQ treated, and because this treatment is not residual, reinfection can occur later in the production cycle (*Power et al., 2019*, *2021*).

This study investigates the infection levels of *Cardicola* spp. in ranched SBT at harvest in July 2018, 2019, and 2021. The analysis focuses on a time frame of 3–5 months after PZQ treatment, considering the variability in SBT condition, ranching duration, PZQ treatment dose, and number of pontoons left untreated each season. This study aims to increase our understanding of the longer-term effects associated with different SBT production and PZQ treatment strategies employed by commercial companies. This is the first study to document *Cardicola* spp. infection in ranched SBT across every commercial company in operation. In addition, studying the epidemiology of *Cardicola* spp. in ranched SBT helps to provide more insights into fluke biology. Samples were collected during commercial operations, which limited the number of sampling time points available for this study.

## MATERIALS AND METHODS

### Ethics statement

All SBT sampled in this study were captured, ranched and euthanised as part of the commercial harvest by aquaculture company personnel in accordance with standards established by the Australian Southern Bluefin Tuna Industry Association. SBT were captured using purse seine nets and towed slowly to grow-out pontoons where they were fed local sardines (*Sardinops sagax*) daily until harvest. SBT were euthanised within 10 s of being out of the water by a spike into their brain, also known as the 'iki jime' method. All work with animals, samples, and methods for recovering samples were approved by RMIT University Animal Ethics Committee (project number 22802) and University of Tasmania Animal Ethics Committee (project number A0016320).

### Sample collection and processing

Wild SBT were captured in the Great Australian Bight during austral summer months and towed to grow-out pontoons located in the Lincoln aquaculture zone of lower Spencer Gulf near Port Lincoln, South Australia (33°27′S, 132°04′E) (Fig. 1). SBT were sampled from

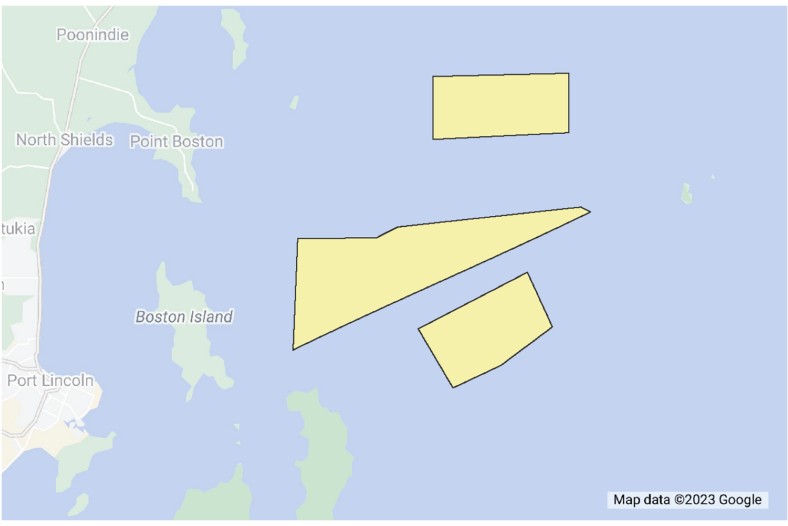

**Figure 1 Location of the Lincoln aquaculture zone (shown in yellow) near Port Lincoln, South Australia where all company lease sites are located for this study (33°27′S, 132°04′E).**

seven commercial companies during harvest operations in July over three ranching seasons (2018, 2019 and 2021). Samples to monitor blood fluke infection could not be collected in 2020 due to Government imposed travel restrictions related to the COVID-19 pandemic. Company leases remained in the same location over the study period.

All companies utilised grow-out pontoons in an area between 20–25 m water depth. PZQ was orally administered (*via* injected baitfish) over 1–2 days and quantities were calculated by Veterinarians using stock and biomass assessments after inspection (K. Rough, 2022, personal communication). All companies treated week 5 of ranching. Three companies (A, B, C) had some untreated pontoons during at least one of the seasons when SBT were sampled and four companies (D–G) treated all pontoons with PZQ every year. However, only SBT from treated pontoons were sampled in this study. One pontoon was sampled per company each sampling year (Table 1). In 2021, Company C and D stocked the same pontoons twice (4 weeks apart for Company C and 6 weeks apart for Company D) and as a result treated SBT twice (5 weeks post stocking), so it is unknown whether SBT sampled from these pontoons were treated once or twice. Date of transfer from tow to grow-out pontoon, treatment date, treatment dose and duration of ranching varied between companies (Table 1).

Weight and fork length for each SBT sampled were collected, and a condition index calculated using the formula: condition index = whole weight (kg)/length (m)$^3$. As SBT were sampled during harvest operations, whole weight for each fish was estimated from the following formula: whole weight = gilled and gutted weight (kg)/0.87 (*Ellis & Kiessling, 2016*). Samples were collected on the vessel during the harvesting process and placed into the relevant buffering agent or on ice within 2–3-min of fish euthanasia, as described (*Power et al., 2019, 2021*). Briefly, small pieces of heart and gill samples (roughly 0.5 cm$^3$) were collected and preserved in RNAlater® (Thermo Fisher Scientific, Scoresby, Victoria,
**Table 1 Company information for sampled Southern Bluefin Tuna including pontoon and treatment characteristics collected from Company A–G in July 2018, 2019 and 2021.**

| Company | Year | Transfer date | Treatment dose (mg/kg) | Duration of ranching (week) | Cumulative mortality (%) |
|---------|------|---------------|------------------------|------------------------------|--------------------------|
| A | 2018 | 28 Feb | 15 | 17 | 0.19 |
|   | 2019 | 7 Mar | 22 | 17 | 0.43 |
|   | 2021 | 1 Apr | 20 | 14 | 0.17 |
| B | 2018 | 28 Mar | 24 | 17 | 0.18 |
|   | 2019 | 27 Mar | 24 | 17 | 0.49 |
|   | 2021 | 14 Mar | 22 | 17 | 0.36 |
| C | 2018 | 10 Mar | 24 | 17 | 0.29 |
|   | 2019 | 1 Apr | 22 | 16 | 0.03 |
|   | 2021* | 14 Mar/8 Apr | 30 | 18/13 | 0.76 |
| D | 2018 | 5 Feb | 20 | 21 | 2.90 |
|   | 2019 | 1 Mar | 20 | 19 | 5.80 |
|   | 2021* | 24 Feb/9 Apr | 18 | 20/14 | 0.38 |
| E | 2018 | 8 Mar | 42 | 17 | 2.33 |
|   | 2019 | 20 Mar | 18 | 17 | 0.27 |
|   | 2021 | 12 Mar | 30 | 17 | 3.00 |
| F | 2018 | 13 Mar | 30 | 17 | 0.56 |
|   | 2019 | 16 Mar | 30 | 18 | 0.24 |
|   | 2021 | 6 Mar | 43 | 18 | 0.11 |
| G | 2018 | 27 Feb | 30 | 20 | 0.08 |
|   | 2019 | 1 Mar | 30 | 20 | 0.52 |
|   | 2021 | 24 Mar | 30 | 16 | 0.54 |

Notes:
Pontoons treated week 5 of ranching. $n$ = 15 each pontoon sampled.
* Pontoon stocked and treated twice.

Australia). In the laboratory each SBT heart was dissected and flushed with tap water to isolate and count adult *C. forsteri*. Gill filaments from the middle region of the second left gill arch were dissected to count *Cardicola* spp. eggs in individual gill filaments, enumerated at 100× magnification using a compound microscope. Eggs were quantified as eggs/mm gill filament length, taking an average from four filaments. DNA was extracted from RNAlater preserved SBT heart and gill samples using the Isolate II Genomic DNA Kit (Bioline) and *C. forsteri* and *C. orientalis* were detected and quantified using quantitative polymerase chain reaction (qPCR) methods described in *Power et al. (2019)*. Briefly, qPCR assays were performed on a Rotor-Gene™ Q (Qiagen, Hilden, Germany) and reaction cycling conditions were: 95 °C for 5 min, followed by 40 cycles of 95 °C for 10 s (denaturing) and 60 °C for 25 s (annealing). All samples tested were analysed in duplicates, including a positive control and no template control for each run. Primers and probes targeting the internal transcribed spacer-2 (ITS-2) region of rDNA to detect *C. forsteri* and *C. orientalis* were designed in previous studies, which confirmed their specificity (*Neumann et al., 2018*; *Polinski et al., 2013*). For *C. forsteri*, forward primer (5′-TGATTGCTTGCTTTTTTCTCGAT-′3), reverse primer (5′-TATCAAAACATCAATCGACATC-3′) and probe (5′-HEX-CCACGACCTGAGCACAAGCCG–BHQ1-3′) were used. For *C. orientalis*, forward primer

(5′-TGCTTGCTATTCCTAGATGTTTAC-3′), reverse primer (5′-AACAACTATACTAA GCCACAA-3′) and probe (5′-HEX–CACAAGCCGCTACCACAATTCCACTC–BHQ1-3′) were used. Cumulative mortalities for the whole ranching season were obtained from companies for each pontoon sampled.

## Statistics

*Cardicola forsteri* and *C. orientalis* infections were described by prevalence (percentage of infected hosts in sampled population) and intensity (average number of adults, eggs, or copy number/mg DNA per infected host) as per *Bush et al. (1997)*. Differences in *Cardicola* spp. infection prevalence by company and year were evaluated using Chi-square (>2 groups) or Fisher's exact test (two groups). Logistic regression was used to determine the relationship between time, SBT condition index and PZQ treatment dose on *Cardicola* spp. infection prevalence. Variables were first analysed individually through univariate regression, and if $p < 0.25$ they were incorporated into multivariate regression models. For intensity, data did not meet the assumptions of normality so non-parametric tests were used. Differences in *Cardicola* spp. infection intensity were evaluated by company and year using Kruskal–Wallis followed by Dunn's (>2 groups) or Mann–Whitney test (two groups). Spearman's rank correlation coefficients were used to determine the relationship between time, SBT condition index and PZQ treatment dose on *Cardicola* spp. infection intensity. The relationship between cumulative mortality and ranching duration/*Cardicola* spp. infection was determined using simple linear regression. Company C and D stocked pontoons with SBT and treated SBT twice in 2021, so data were removed from regression and correlation analyses as duration of ranching could not be determined for sampled fish. All statistical analyses were performed using GraphPad Prism 8 (GraphPad software; GraphPad, San Diego, CA, USA). Significance for all statistical analyses was assumed at $p \leq 0.05$.

## RESULTS

### Condition index analysis

Mean SBT condition index was significantly different between companies in 2018 (H = 24.54, $p < 0.0001$) and 2021 (H = 50.90, $p < 0.0001$) (Fig. 2). In 2018, mean condition index was significantly higher for Company B than Company C ($p = 0.0060$), Company D ($p = 0.0006$) and Company F ($p = 0.0117$); Company E was significantly higher than Company D ($p = 0.0130$). In 2021, mean condition index for Company G was significantly lower than Company A ($p = 0.0055$), Company B ($p = 0.0002$), Company D ($p < 0.0001$), Company E ($p < 0.0001$) and Company F ($p < 0.0001$); Company D was significantly higher than Company C ($p = 0.0183$). No differences were seen between companies in 2019. Data were not obtained for Company A and Company G in 2018.

No significant correlation was seen between SBT condition index and weeks in ranching at time of harvest (Spearman's r = 0.0808, d.f. = 241, $p = 0.2112$).

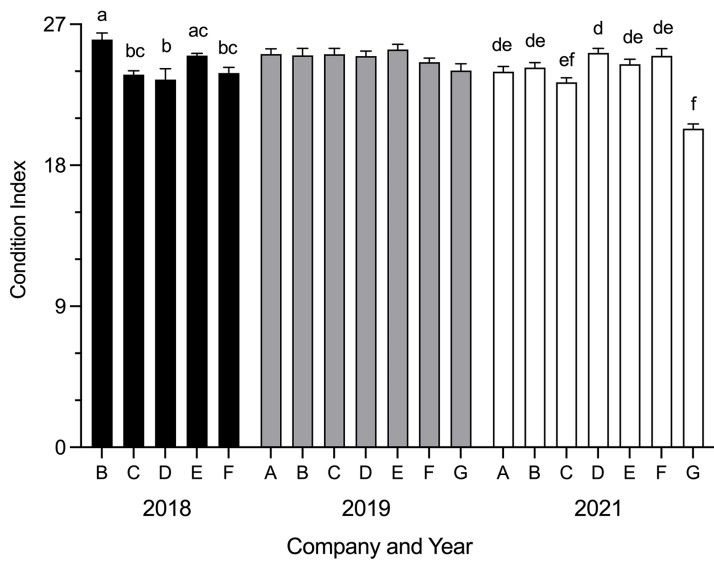

**Figure 2 Mean SBT condition index (±SE) from Company A–G PZQ treated pontoons at harvest in July 2018, 2019 and 2021 ($n$ = 12–15 for each company).** Condition index data not collected from Company A and Company G in 2018. Different letters denote statistical differences at $p \leq 0.05$ between companies for each year.

## Cumulative mortality

Cumulative mortality was highest in a pontoon sampled from Company D in 2018 (2.90%) and 2019 (5.80%), and in 2021 it was highest in a pontoon sampled from Company E (3.00%) (Table 1). There was no relationship between cumulative mortality and ranching duration for SBT pontoons sampled in this study (d.f. = 15, $R^2$ = 0.001, $p$ = 0.8889). No relationship was seen between cumulative mortality and intensity of *Cardicola* spp. infection from any diagnostic method (adult *C. forsteri* d.f. = 15, $R^2$ = 0.091, $p$ = 0.2097; *C. forsteri* (ITS-2) in heart d.f. = 15, $R^2$ = 0.022, $p$ = 0.5427; *Cardicola* spp. eggs in gill filament d.f. = 15, $R^2$ = 0.002, $p$ = 0.8632; *C. forsteri* (ITS-2) in gills d.f. = 15, $R^2$ = 0.001, $p$ = 0.9486; *C. orientalis* (ITS-2) in gills d.f. = 15, $R^2$ = 0.031, $p$ = 0.4685).

## Blood fluke prevalence

Four variables (time—number of weeks post PZQ treatment, time—year, SBT condition index and PZQ treatment dose) were examined for their effect on prevalence of *Cardicola* spp. infection in treated SBT using each diagnostic method (Tables 2 and 3). Time after PZQ treatment (number of weeks) increased the likelihood of infection based on adult *C. forsteri* in heart (OR = 1.206, $p$ = 0.0122) and *Cardicola* spp. DNA detected in gills (positive qPCR of ITS-2) (OR = 1.403, $p$ = 0.0004). Time (by year) decreased the likelihood of infection based on *Cardicola* spp. eggs in gills (OR = 0.627, $p$ = 0.0452). SBT condition index and PZQ treatment dose showed no significant effect on prevalence of *Cardicola* spp. infection. No multicollinearity was seen in any of the logistic regressions and all models fit the data as the goodness of fit (Hosmer–Lemeshow test) was not significant (adult *C. forsteri* in heart χ2 = 11.62, $p$ = 0.1690; *C. forsteri* (ITS-2) in heart χ2 = 4.311 $p$ = 0.1159;
**Table 2 Variables examined for their effect on prevalence of *Cardicola* spp. infection through simple logistic regression (OR, odds ratio; Z, regression coefficient).**

|  | Adult *C. forsteri* in heart | | | *C. forsteri* (ITS-2) in heart | | | *Cardicola* spp. eggs in gills | | | *C. forsteri* and *C. orientalis* (ITS-2) in gills | | |
|---|---|---|---|---|---|---|---|---|---|---|---|---|
|  | OR | Z | $p$ | OR | Z | $p$ | OR | Z | $p$ | OR | Z | $p$ |
| Time (weeks post PZQ treatment) | 1.213 | 2.585 | 0.0097 | 1.135 | 1.596 | 0.1104 | 1.379 | 2.872 | 0.0041 | 1.467 | 4.195 | <0.0001 |
| Time (year) | 0.933 | 0.664 | 0.5061 | 1.133 | 1.132 | 0.2576 | 0.748 | 2.312 | 0.0216 | 0.727 | 3.037 | 0.0022 |
| SBT condition index | 1.083 | 1.079 | 0.2807 | 0.952 | 0.660 | 0.5093 | 1.006 | 0.071 | 0.9430 | 1.000 | 0.001 | 0.9995 |
| Treatment dose | 1.025 | 1.492 | 01357 | 0.932 | 1.376 | 0.1690 | 1.453 | 1.793 | 0.0729 | 1.382 | 0.058 | 0.9539 |

**Table 3 Variables examined for their effect on prevalence of *Cardicola* spp. infection through multiple logistic regression (OR, odds ratio; Z, regression coefficient).**

|  | Adult *C. forsteri* in heart | | | *C. forsteri* (ITS-2) in heart | | | *Cardicola* spp. eggs in gills | | | *C. forsteri* and *C. orientalis* (ITS-2) in gills | | |
|---|---|---|---|---|---|---|---|---|---|---|---|---|
|  | OR | Z | $p$ | OR | Z | $p$ | OR | Z | $p$ | OR | Z | $p$ |
| Time (weeks post PZQ treatment) | 1.206 | 2.506 | 0.0122 | 1.126 | 1.513 | 0.1304 | 1.207 | 1.583 | 0.1135 | 1.403 | 3.529 | 0.0004 |
| Time (year) | n.a | n.a | n.a | n.a | n.a | n.a | 0.627 | 2.002 | 0.0452 | 0.7552 | 1.656 | 0.0977 |
| Treatment dose | 1.023 | 1.330 | 0.1836 | 1.024 | 1.276 | 0.2019 | 1.047 | 1.798 | 0.0722 | n.a | n.a | n.a |

*Cardicola* spp. eggs in gills $\chi 2 = 5.851$, $p = 0.6640$; *C. forsteri* and *C. orientalis* (ITS-2) in gills $\chi 2 = 7.417$, $p = 0.3868$).

No statistical differences in prevalence of *C. orientalis* (based on positive qPCR of ITS-2) in gill samples was seen. *Cardicola orientalis* (ITS-2) was detected in one sample from Company A in 2018 (prevalence 7.14%, CI [0.37–31.5%]), one sample from Company B in 2018 (prevalence 6.67% CI [0.34–29.8%]), one sample from Company D in 2019 (prevalence 6.67%, CI [0.34–29.8%]), and two samples from Company G in 2019 (prevalence 13.3%, CI [2.37–37.9%]). *Cardicola orientalis* (ITS-2) was not detected in any gill samples in 2021.

Year and company were analysed for their effect on prevalence of *Cardicola* spp. infection in treated SBT using each diagnostic method (Fig. 3, see Table S1 for 95% CI and Tables S2 and S3 for statistical values). To summarise, when comparing companies each year, Company A had the highest prevalence for every diagnostic method in 2018, and Company G had the highest prevalence for every diagnostic method in 2019. No consistent pattern was seen between companies in 2021. When comparing the same company between years, prevalence decreased over time for Company A, C and D for every diagnostic method. High variance in prevalence was seen between years for Company B, E, F and G. When comparing pontoons from companies which treated once (Company A, B, E, F, G) and twice (Company C and D) in 2021, prevalence of adult *C. forsteri* was lower for Company C and D than Company B, E and F but there was no difference between Company A or G and C or D. Prevalence of *C. forsteri* (based on positive qPCR of ITS-2) in

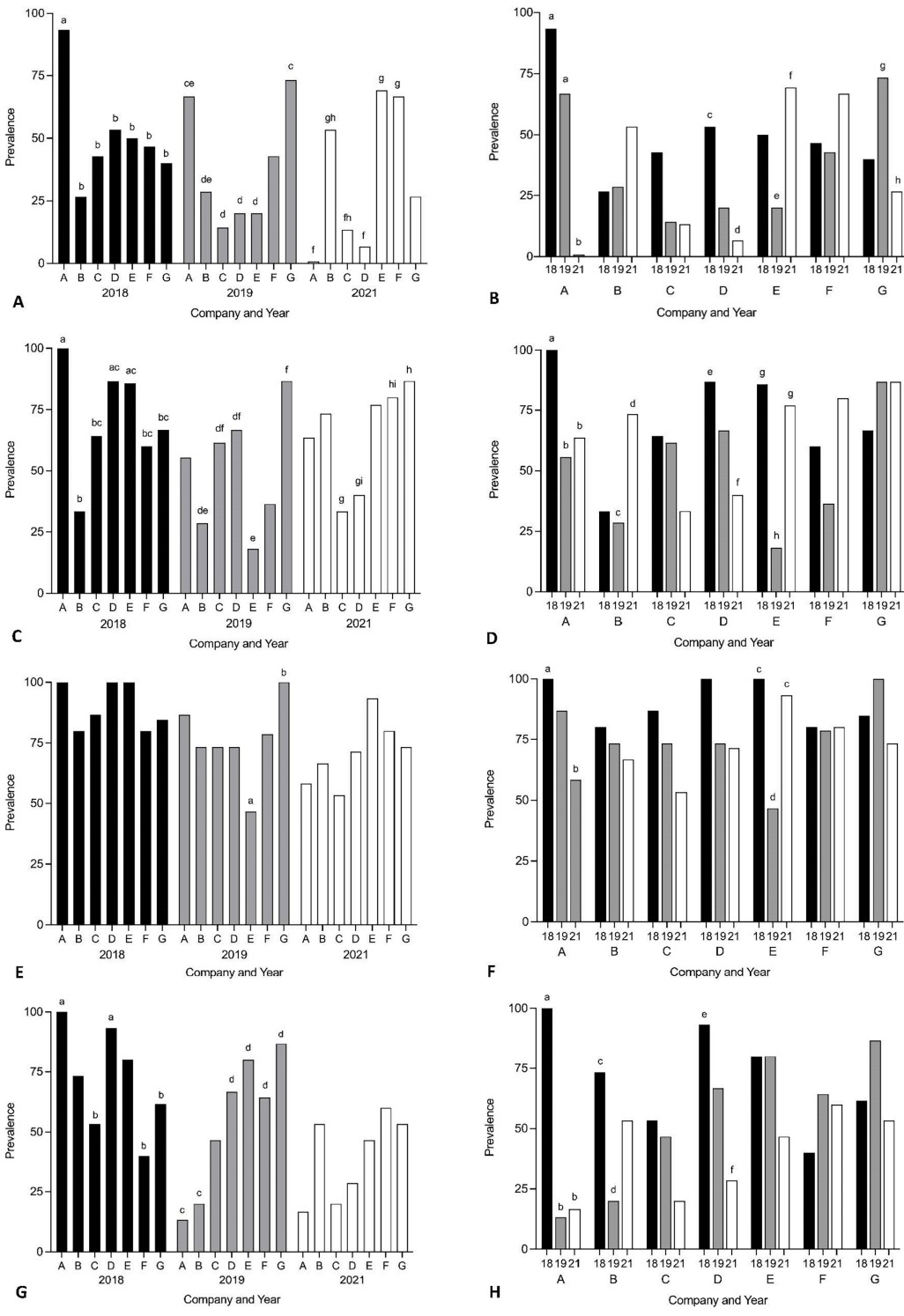

**Figure 3 Prevalence of *Cardicola* spp. infection in SBT from Company A–G PZQ treated pontoons in July 2018, 2019 and 2021 (*n* = 12–15 for each company).** (A and B) Prevalence of adult *Cardicola forsteri* infection in heart (microscopy); (C and D) prevalence of *Cardicola forsteri* (qPCR, ITS-2) in heart; (E and F) prevalence of *Cardicola* spp. eggs in gill filament (microscopy); (G and H) prevalence of *Cardicola forsteri* (qPCR, ITS-2) in gills. Different letters denote statistical differences at $p \leq 0.05$ between companies for each year (A, C, E, G) and between years for each company (B, D, F, H).

**Table 4 Spearman's correlation between SBT/pontoon characteristics and *Cardicola* spp. infection intensity in SBT from PZQ treated pontoons sampled in July 2018, 2019 and 2021.**

| | Adult *C. forsteri* in heart | | | *C. forsteri* (ITS-2) copy number/mg in heart | | | *Cardicola* spp. eggs in gills | | | *C. forsteri* and *C. orientalis* (ITS-2) in gills | | |
|---|---|---|---|---|---|---|---|---|---|---|---|---|
| | r | d.f. | p | r | d.f. | p | r | d.f. | p | r | d.f. | p |
| Time (weeks post PZQ treatment) | 0.0743 | 122 | 0.4161 | 0.0597 | 167 | 0.4436 | 0.0593 | 226 | 0.3746 | 0.1136 | 163 | 0.1489 |
| Time (year) | −0.0687 | 122 | 0.4522 | 0.1231 | 167 | 0.1130 | −0.0545 | 226 | 0.4149 | 0.0284 | 163 | 0.7194 |
| SBT condition index | 0.0833 | 94 | 0.4248 | −0.0583 | 140 | 0.4939 | −0.0299 | 191 | 0.6812 | −0.2426 | 138 | 0.0041 |
| PZQ treatment dose | −0.0166 | 122 | 0.8563 | −0.3553 | 167 | <0.0001 | −0.1136 | 226 | 0.0883 | −0.0265 | 163 | 0.7368 |

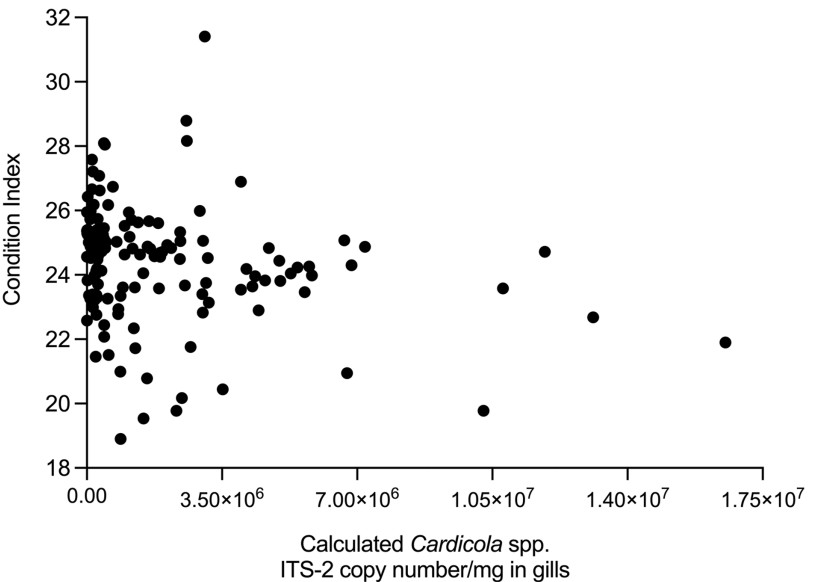

**Figure 4 Correlation between SBT Condition Index and intensity of calculated *Cardicola forsteri* and *C. orientalis* ITS-2 copy number/mg in SBT gills (Spearman's r = −0.2426, d.f. = 138, p = 0.0041).**

heart samples was lower for Company C and D than Company F and G. When comparing prevalence from companies which utilised untreated pontoons in addition to the sampled treated pontoons (Company A, B, C) and companies which did not (Company D–G), no consistent pattern could be seen between companies each year.

## Blood fluke intensity

The relationship between time, SBT condition index and PZQ treatment dose on *Cardicola* spp. infection intensity in treated SBT was analysed using each diagnostic method (Table 4). A statistically significant negative correlation was seen between SBT Condition Index and intensity of calculated *Cardicola forsteri* and *C. orientalis* ITS-2 copy number/ mg in SBT gills (Spearman's r = −0.2426, d.f. = 138, p = 0.0041) (Fig. 4). A statistically significant negative correlation was seen between PZQ treatment dose and intensity of

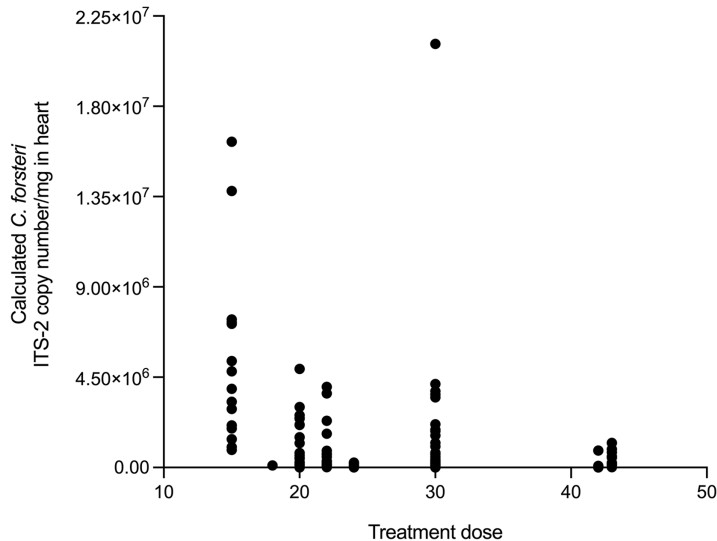

**Figure 5 Correlation between PZQ treatment dose and intensity of calculated *Cardicola forsteri* ITS-2 copy number/mg in SBT heart (Spearman's r = −0.3553, d.f. = 167, *p* < 0.0001).**

calculated *Cardicola forsteri* ITS-2 copy number/mg in SBT heart (Spearman's r = −0.3553, d.f. = 167, *p* < 0.0001) (Fig. 5).

Year and company were analysed for their effect on intensity of *Cardicola* spp. infection in treated SBT using each diagnostic method (Fig. 6, see Table S4 for mean intensity ± SE values and Tables S5 and S6 for all statistical values). To summarise, Company A had the highest mean intensity for every diagnostic method in 2018, and Company G had the highest mean intensity for every diagnostic method in 2019. No consistent pattern was seen between companies in 2021. When comparing mean intensity from the same company between years, intensity was highest in 2018 for Company A across all diagnostic methods. There was high variability for other companies with no consistent trend seen between years. When comparing companies which treated once (Company A, B, E, F, G) and those that treated twice (Company C and D) in 2021, no differences were seen in mean intensity for any diagnostic method. When comparing mean intensity from companies which utilised untreated pontoons in addition to the sampled treated pontoons (Company A, B, C) and companies which did not (Company D–G), no consistent pattern was seen between companies each year. No statistical differences in mean intensity of *C. orientalis* (based on positive qPCR of ITS-2) in gill samples was seen.

## DISCUSSION

A statistically significant negative relationship was seen between Condition Index and intensity of *Cardicola* spp. DNA in SBT gills, potentially demonstrating blood fluke infection has a negative effect on SBT growth. The effects of blood flukes *Sanguinicola inermis* on farmed carp, *Cyprinus carpio*, and *C. klamathensis* on cutthroat trout, *Oncorhynchus clarkia*, showed poor growth performance with depressed growth rates and daily live weight gain (*Evans, 1974*; *Iqbal & Sommerville, 1986*). Poor growth rates are

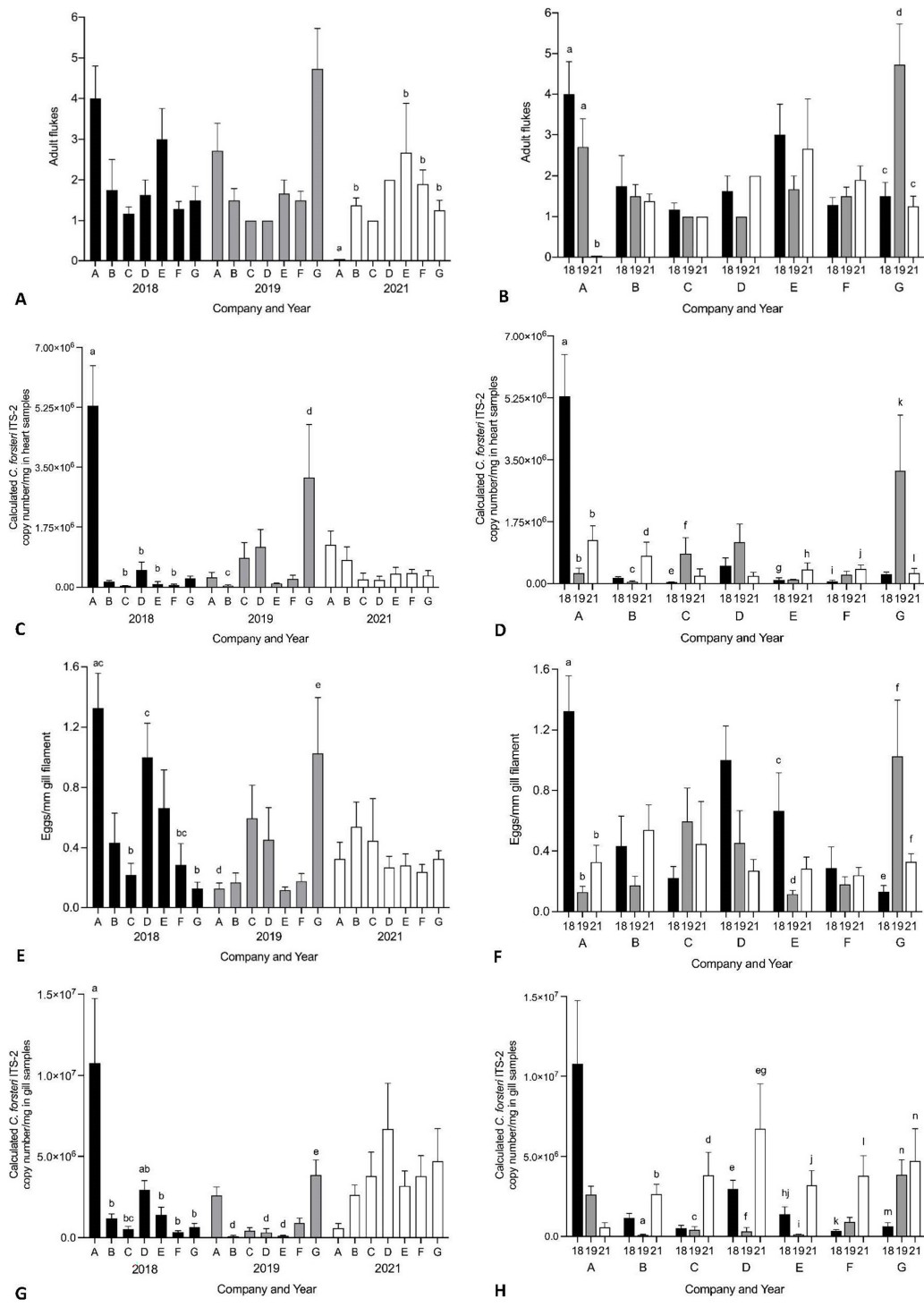

**Figure 6 Mean intensity of *Cardicola* spp. infection in SBT from Company A–G PZQ treated pontoons in July 2018, 2019 and 2021 (*n* = 12–15 for each company).** (A and B) Mean intensity (±SE) of adult *Cardicola forsteri* infection in heart (microscopy); (C and D) mean intensity (±SE) of *Cardicola forsteri* (qPCR, ITS-2) in heart; (E and F) mean intensity (±SE) of *Cardicola* spp. eggs in gill filament (microscopy); (G and H) mean intensity (±SE) of *Cardicola forsteri* (qPCR, ITS-2) in gills. Different letters denote statistical differences at *p* ≤ 0.05 between companies for each year (A, C, E, G) and between years for each company (B, D, F, H).

thought to be the result of decreased food intake and reduced feeding was a clinical feature noted in both experiments from infected fish. Growth suppression was more pronounced in heavily infected fish but also occurred in fish with lower infection levels when compared to uninfected carp fed the same diet (*Iqbal & Sommerville, 1986*). No relationship was seen between Condition Index and infection intensity in ranched SBT sampled between 2004–2006 but infection was only measured by the number of adult flukes in heart and not by the number of eggs in gills (*Aiken, Hayward & Nowak, 2015*). Accumulation of blood fluke eggs is thought to be the main source of pathogenesis in bluefin tuna, so this is more likely to account for suppressed SBT growth (*Shirakashi et al., 2012b*). Experimental work to demonstrate *Cardicola* spp. infection, SBT food intake, and SBT growth rates would be ideal, however SBT are difficult to study experimentally due to their size and conservation status (*Ellis & Kiessling, 2016*). As there is no SBT hatchery it is impossible to obtain smaller individuals for experiments. Infection levels observed in this study, whilst most likely too low to have any apparent association to cumulative mortalities, could still result in sublethal effects such as decreased growth in ranched SBT.

Epidemiological results from this study indicate PZQ remains effective at controlling *Cardicola* spp. infection in ranched SBT, 10 years after PZQ administration began for some commercial companies. Maximum mean intensity of *C. forsteri* in SBT heart was 4.73 adults, similar to intensities seen before the introduction of PZQ treatment at harvest when no gross pathology or mortalities were attributed to *C. forsteri* infection, and markedly lower than intensities of 40–270 adults at 2–3 months of ranching when mortality epidemics were recorded before PZQ treatment began (*Aiken, Hayward & Nowak, 2006*; *Hayward et al., 2010*). No increase in prevalence or intensity of *Cardicola* spp. infection was seen from 2018 to 2021 overall for any diagnostic method, indicating that PZQ has maintained its effectiveness. A higher PZQ treatment dose was negatively correlated with intensity of calculated *C. forsteri* (ITS-2) in heart. This could mean a higher treatment dose is more likely to kill adult *C. forsteri*, helping to reduce intensity of calculated *C. forsteri* (ITS-2) in heart. Given treatment dose did not have an impact on *Cardicola* spp. prevalence or intensity as shown by other diagnostic methods, more research is required to determine if this is causal. PZQ has been used widely to treat flatworm infections for over 40 years, most notably schistosomiasis (*Norbury et al., 2022*). Although schistosomes with reduced sensitivity to PZQ have been generated in the laboratory and reduced efficacy has been reported in PZQ treatment of schistosomiasis, widespread resistance to PZQ has not yet emerged (*Greenberg & Doenhoff, 2017*). In Norway, where PZQ is used as an oral treatment for tapeworm infections (*Eubothrium* sp.) in sea farmed Atlantic Salmon, *Salmo salar* L., treatment failure and concerns regarding development of resistance have been reported (*Hjeltnes et al., 2018*, *2019*). No good alternative to PZQ has been found to date in Norway, and the salmon industry is required to find new control methods to optimally treat or reduce *Eubothrium* sp. infections (*Geitung et al., 2021*). Control of *Cardicola* spp. infection in SBT is currently highly dependent on the continued efficacy of PZQ, so ongoing monitoring will be important to evaluate overall effectiveness of PZQ as a control measure over time, particularly regarding different treatment doses and duration of

ranching. Once other alternatives to PZQ are developed, their potential application in SBT ranching should be investigated.

No consistent pattern of *Cardicola* spp. infection was seen at harvest based on company. Variability was seen between years, with some companies increasing over time, some decreasing, and some showing no pattern. Company A in 2018 and Company G in 2019 had higher prevalence and intensity of *Cardicola* spp. infection compared to other companies in the same year across most diagnostic methods. However, this did not correspond to higher rates of cumulative mortality or lowered condition of SBT as seen by condition index. Although companies utilised the same lease site across ranching seasons in this study, changes in pontoon location each ranching season could impact prevalence of the intermediate host and the likelihood of vertical dispersal of *Cardicola* spp. by water depth and currents, so this may account for the variation seen between years. As no samples were collected prior to ranching, variability of wild SBT caught by different companies is unknown and may also affect results (*Aiken, Hayward & Nowak, 2015*). The likelihood of blood fluke infection decreased over time (in years) in SBT gills, but some companies harvested earlier in 2021 than previous years and some could not be included in analysis given they stocked and treated pontoons twice, so this may account for the decrease. It is also possible that consistent years of PZQ treatment have lowered the parasite load in the environment, making reinfection after treatment less likely, but more years of epidemiological data would be needed to determine this.

Economic considerations have led some companies to leave some of their pontoons untreated. In Australia, 0.02 mg/kg has been set as the maximum residue limit for PZQ in fish muscle, so a withholding period is necessary before SBT are harvested (*Norbury et al., 2022*). There are also costs associated with PZQ treatment. The absence of PZQ treatment can have an effect on blood fluke intensity, however infection intensity recorded in untreated pontoons during the 2018 and 2019 ranching season was too low to cause any significant decrease in the condition of SBT or increase mortalities (*Power et al., 2019, 2021*). When comparing *Cardicola* spp. infections in treated pontoons between companies that utilised untreated pontoons (Company A–C) and those that didn't (Company D–G), no consistent differences in prevalence or intensity could be seen in this study. This analysis supports evidence that horizontal dispersal of *Cardicola* spp. is less of a concern than vertical dispersal given the free-living cercarial stage is very small with a short, simple tail they are likely poor swimmers and rely on water currents for dispersal (*Cribb et al., 2011*; *Shirakashi et al., 2016*). The addition of untreated pontoons did not appear to have an effect on *Cardicola* spp. infection for treated pontoons in operation nearby.

Reinfection of *Cardicola* spp. in SBT was expected post treatment given the assumed proximity of intermediate host/s to SBT pontoons, PZQ is not efficacious against early life stages *e.g.*, eggs or miracidia, and treatment is not residual (*Hardy-Smith et al., 2012*; *Shirakashi et al., 2012a*). Prior to the introduction of PZQ treatment, prevalence and intensity of *Cardicola* spp. increased rapidly during ranching and peaked 2–3 months post transfer before tapering off (*Aiken, Hayward & Nowak, 2006, 2015*), so the timing of single treatment application at week 5 enables the industry to control *Cardicola* spp. infections before SBT are harvested. Samples from this study were collected between week 14–21 of

ranching (9–16 weeks post treatment), and the likelihood of infection increased over ranching time for adult *C. forsteri* in heart and *Cardicola* spp. (based on positive qPCR of ITS-2 rDNA) in gills. Given that all companies treated week 5, this is likely the effect of time since PZQ treatment. The same increase was not seen for intensity of infection, so the same rapid increase and tapering off post treatment as seen during seasons before the introduction of PZQ treatment was absent here. This could be an indication that SBT may have developed some level of resistance against reinfection. SBT ranched over two seasons had a higher antibody response and lower blood fluke prevalence and abundance in the second season than SBT ranched in a single season, indicating some level of immunity most likely as a result of previous infection (*Aiken et al., 2008*). Antibody levels should be investigated in conjunction with blood fluke prevalence and intensity to determine development and duration of SBT resistance against *Cardicola* spp. reinfection post treatment.

Prevalence of *C. orientalis* in this study was very low; only detected in a few samples in 2018 and 2019, and not detected at all in 2021. *Cardicola orientalis* DNA was detected in 86% of samples collected between 2008–2012 (*Polinski et al., 2013*), but has rarely been detected since (*Neumann et al., 2018*; *Power et al., 2019*, *2021*). It is likely that the wide-ranging geographical distribution of *Cardicola* spp. is due to overlapping migratory patterns of bluefin tuna (*Aiken et al., 2007*). Juvenile SBT undertake seasonal migrations, aggregating in the Great Australian Bight during the austral summer then dispersing east and west during the autumn (*Patterson et al., 2018*). SBT migrating west may overlap with migratory patterns of Atlantic Bluefin Tuna (ABT), *T. thynnus*, before returning to Australian waters (*Arrizabalaga et al., 2015*). If so, they may be more likely to encounter the intermediate host or hosts of *C. orientalis* when cercariae are being released (*Shirakashi et al., 2016*). Less is known about ABT from the South Atlantic Ocean as no major commercial fishery exists there, but there is evidence to suggest a return of ABT in parts of the North Atlantic Ocean from 2012 onwards (*Nøttestad, Boge & Ferter, 2020*; *Horton et al., 2021*). It is possible that there was more overlap between ABT and SBT migrations during 2008–2012 when a higher prevalence of *C. orientalis* DNA was seen. The only known intermediate host for *C. orientalis* is terebellid polychaete *Nicolea gracilibranchis* from the Pacific Ocean, Japan, which is known to have a cosmopolitan distribution (*Day, 1973*; *Shirakashi et al., 2016*). However, the wide-ranging distribution of the intermediate host or hosts of *C. orientalis* are unknown, and no resounding evidence for the change in *Cardicola* species dynamic has been found to date.

## CONCLUSIONS

This study documents *Cardicola* spp. infection in ranched SBT from all commercial companies during 2018, 2019 and 2021 ranching seasons. This is the most comprehensive survey of blood fluke infection in ranched SBT to date. SBT Condition Index decreased as intensity of *Cardicola* spp. DNA in SBT gills increased, suggesting blood fluke infection has a negative effect on SBT growth. PZQ remains effective at controlling *Cardicola* spp. infection in ranched SBT. *Cardicola forsteri* was the dominant species detected, and

*C. orientalis* rarely detected. Ongoing monitoring is important for continued vigilance of PZQ efficacy, especially while alternative treatment measures remain absent.

## ACKNOWLEDGEMENTS

The authors thank the participating commercial companies involved in this research. We are grateful to researchers Dr Daniel Huston and Dr Jeremiah Minich for their help with field work in Port Lincoln.

### Funding

This work was supported by Fisheries Research and Development Corporation (FRDC2017-241 and FRDC2018-170). The funders had no role in study design, data collection and analysis, decision to publish, or preparation of the manuscript.

### Grant Disclosures

The following grant information was disclosed by the authors:
Fisheries Research and Development Corporation: FRDC2017-241 and FRDC2018-170.

### Competing Interests

Barbara F. Nowak is an Academic Editor for PeerJ. Kirsten Rough is an employee of the Australian Southern Bluefin Tuna Industry Association.

### Author Contributions

- Cecilia Power conceived and designed the experiments, performed the experiments, analyzed the data, prepared figures and/or tables, authored or reviewed drafts of the article, and approved the final draft.
- Melissa Carabott performed the experiments, authored or reviewed drafts of the article, and approved the final draft.
- Maree Widdicombe performed the experiments, prepared figures and/or tables, authored or reviewed drafts of the article, and approved the final draft.
- Lachlan Coff performed the experiments, authored or reviewed drafts of the article, and approved the final draft.
- Kirsten Rough conceived and designed the experiments, authored or reviewed drafts of the article, and approved the final draft.
- Barbara Nowak conceived and designed the experiments, authored or reviewed drafts of the article, and approved the final draft.
- Nathan J. Bott conceived and designed the experiments, authored or reviewed drafts of the article, and approved the final draft.

### Animal Ethics

The following information was supplied relating to ethical approvals (*i.e.*, approving body and any reference numbers):

RMIT Animal Ethics Committee (22802) and UTAS Research Ethics and Integrity Unit (A0016320) provided approval for this research. All animals were harvested and euthanised as part of the commercial harvest by aquaculture company personnel in accordance with standards established by the Australian Southern Bluefin Tuna Association.

## Data Availability

The raw data are available in the Supplemental Files.

## Supplemental Information

Supplemental information for this article can be found online at http://dx.doi.org/10.7717/peerj.15763#supplemental-information.

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
