# Peer review of "Effects of company and season on blood fluke (Cardicola spp.) infection in ranched Southern Bluefin Tuna: preliminary evidence infection has a negative effect on fish growth"

_PeerJ, doi:10.7717/peerj.15763_

## Round 0.1 · original submission · Minor Revisions

I agree with the reviewers that the current version of the manuscript only warrants minor revisions. Kindly prepare a point-to-point rebuttal addressing each concern and I look forward to reading the revised version of the manuscript!

Reviewer 1 ·

Basic reporting

No major issues; the article is well-structured (I suggest rewriting the Results so that long paragraphs with numbers are put in tables; see my comments under 4).

Experimental design

Experimental design is not the ideal, but given the demanding specs and issues with pandemic, it is still robust enough.

Validity of the findings

Statistics has been appropriately used to test the findings validity.

Additional comments

Please add in the abstract and conclusions what was the treatment concentration/ strategy that show the best results.
Introduction: rephrase the last paragraph so the aims are explicitly stated.
Results: listing all the numbers in the text makes the section dull for readers. Please put the statistics in the tables highlighting the differences, and mention in the text only the most prominent or unexpected findings.
Discussion: could authors please add what is the estimated price of PZQ treatment over the ranching period, and what are legal norms when it comes to PZQ clearance from the tissues.

Reviewer 2 ·

Basic reporting

This paper contains useful information on blood fluke (Cardicola spp.) in ranched SBT and, therefore, I think it should be published.


The article has been written in professional English throughout. However, some areas - such as Results > Blood Fluke Prevalence & Blood Fluke Intensity, in my opinion, are too verbose and should be simplified. I hope the authors can consider this recommendations.

Experimental design

Other comments:

Line 72-74:
In regards to SBT capture mechanisms amongst company, were the wild SBT captured at the same period of time for all companies?It would be better for the readers to know their time of capture, perhaps in specific month/season.

Line 85-86:
Does this mean the previously captured SBT in the pontoons were treated twice?How about the newly stocked SBT to this pontoons - were they also treated twice?Please indicate how the PZQ treatment were administered for the new SBT stock - and what is the time gap between 1st & 2nd treatments. And also, clarify how to differentiate each SBT individuals.

Validity of the findings

The data on which the conclusions are based is acceptable and statistically sound, and tested.

Other comments:

Line 139:
Should the highest cumulative mortality in 2018 be Company D (2.9%)? As indicated in Table 1. Please correct accordingly and this might influence the discussion part as well.

Additional comments

No additional comments

Reviewer 3 ·

Basic reporting

The introduction is clear and well established. Method section is also clear and enough to describe. However, results and discussion section require revision prior to publication.

Experimental design

A point that I should remain comment is a method of qPCR (Line 99- 105). The authors explained by using previous reports, thus potential readers have to check them out. It is better to state at least outline of qPCR method such as number of cycles, temperature and primer IDs (accession number, date base etc.).
I have a question which should be mentioned in text that the authors have not described enough information for the feeding condition of each company (line 66). What kind of baitfish did the companies use for SBT? And did these types/amount of different baitfish influence on growth?

Validity of the findings

The topic of this study is worthy researched and within the interest of potential readers who are in relation to tuna farmers and biologist of ichthyology. Novel findings are provided by the authors and conclusions are well stated.

Additional comments

It is better to discuss more about what the factors brought these differences between companies in prevalence of Cardicola spp.
Also, if it is possible, the authors should mention that why the growth performance was inferior when fish infected.

Reviewer 4 ·

Basic reporting

The manuscript is well written, with sufficient literature and background context.

The authors could better elaborate on the details of Figure 1. While the integrity of the companies have to be kept secret. the authors could better describe the environmental parameters better. I have provided some comments in the pdf.

Experimental design

No comment.

Validity of the findings

The findings are novel, however the results paragraph are too lengthy. Although the authors have later summarized in a separate paragraph of the significant of the results (example lines 304-314), perhaps highlighting key points (example top 3 companies) based on the already provided figures would suffice.

Additional comments

please address comments from the pdf

Annotated reviews are not available for download in order to protect the identity of reviewers who chose to remain anonymous.

---

## Round 0.2 · accepted · Accept

The authors have addressed all of the concerns raised by the reviewers. After careful review, I believe that the current version of the manuscript is ready for publication.